# Effects of Hypolipidemic Drugs on Psoriasis

**DOI:** 10.3390/metabo13040493

**Published:** 2023-03-29

**Authors:** Mateusz Matwiejuk, Hanna Mysliwiec, Olivia Jakubowicz-Zalewska, Adrian Chabowski, Iwona Flisiak

**Affiliations:** 1Department of Dermatology and Venereology, Medical University of Bialystok, 15-540 Bialystok, Poland; 2Department of Physiology, Medical University of Bialystok, 15-222 Bialystok, Poland

**Keywords:** psoriasis, statins, fibrates, glitazones, PCSK9 inhibitors, GLP-1 analogs

## Abstract

Psoriasis is a chronic, systematic, inflammatory disease in which multiple metabolic and immunologic disturbances lead to lipid abnormalities, impaired glucose tolerance, metabolic syndrome, diabetes mellitus, atherosclerosis, hypertension, ischemic heart disease, and numerous metabolic disorders. In clinical practice, the most commonly used drugs in the treatment of lipid abnormalities are statins and fibrates. Statins are characterized by pleiotropic effects such as antioxidant, anti-inflammatory, anticoagulant, and antiproliferative. They work by reducing the concentrations of low-density lipoprotein (LDL), total cholesterol, and triglycerides and stabilizing atherosclerotic plaque. Fibrates are medications, which help to lower triglycerides, LDL, very low-density lipoprotein (VLDL) levels and increase lower high-density lipoprotein (HDL). In recent years, many new drugs were found to normalize the lipid profile in patients with psoriasis: glitazones (pioglitazone, troglitazone), and glucagon-like peptide-1 (GLP-1) receptor agonists. Pioglitazone improves the lipid profile, including the decrease of triglycerides, fatty acids, and LDL, as well as the increase of HDL. Glucagon-like peptide 1 (GLP-1) analogs decrease modestly low-density lipoprotein cholesterol (LDL-C), total cholesterol, and triglycerides. The purpose of this study is to assess the current state of knowledge on the effect of different hypolipidemic treatments on the course of psoriasis. The study includes literature from medical databases PubMed and Google Scholar. We were browsing PubMed and Google Scholar until the beginning of December. The systematic review includes 41 eligible original articles.

## 1. Introduction

Trong et al. [1] inform that psoriasis is a chronic inflammatory skin disease which prevalence ranges from 1–3% worldwide. It is highlighted that dyslipidemia is one of the most important comorbidities of psoriasis. Dyslipidemia in psoriatic patients (PsO) is characterized by hypertriglyceridemia, hypercholesterolemia, increased LDL-C, and lower HDL. Dyslipidemia is significantly associated with the metabolic syndrome and the risk of death from cardio-vascular diseases. On one hand, the treatment of psoriasis can positively affect the lipid metabolism, but on the other hand, it turns out that the treatment of lipid disorders can have a beneficial effect on the course of psoriasis [1]. Imamura et al. [2] inform us that hypertriglyceridemia may play a serious role in the pathogenesis of psoriasis. The researcher demonstrates that during clofibrate treatment, a significant improvement both in dyslipidemia abnormalities and in psoriatic lesions was noted [2]. Singh et al. [3] mark that in psoriasis, the risk of metabolic syndrome is increased in comparison to healthy patients [3]. Xu et al. [4] report that after a weeks-long procedure of treatment with the glucagon-like peptide (GLP)-1 analog (liraglutide), levels of LDL-C and the psoriasis area and severity index (PASI) were much reduced in PsO [4].

## 2. Materials and Methods

This systematic review was written following the 2020 updated Preferred Reporting Items for Systematic Reviews and Meta-Analyses (PRISMA) guidelines [5,6]. The review protocol was submitted to the Protocols database (registration number 79315).

A medical literature search of PubMed (1991–present) and Google Scholar (1988-present), conducted in the winter of 2022/2023, was performed using appropriate terms without date limitations. The main subject of the research was to identify the effect of hypolipidemic drugs on psoriasis. Medical subject headline terms included “psoriasis and statins”, “psoriasis and fibrates”, “psoriasis and glitazones”, and “psoriasis and GLP-1 analogs”.

Non-English publications, duplicated publications, and articles with low clinical significance, were all excluded from the analysis. Originally, human studies were included in the systematic review. In view of the fact that there are limited data on glitazones and GLP-1 analogs, we have also included some important case reports to the review. The results of the search strings were merged together, and duplicates were removed. Afterwards, the titles and abstracts of the remaining studies were screened in order to identify relevant articles that addressed the review subject. Afterwards, the titles and abstracts of the remaining studies were independently screened by two reviewers (M.M. and H.M.) in order to identify relevant articles that addressed the review subject. Disagreements between reviewers were resolved by the opinion of a third reviewer (A.C.). Finally, the selected eligible articles were fully reviewed.

## 3. Results and Discussion

According to Figure 1, the search resulted in the retrieval of 622 records, of which 102 were screened for relevance and 41 ultimately included in the qualitative synthesis.

### 3.1. Statins’ Influence on Psoriasis

#### 3.1.1. Statins’ Influence on Psoriasis with Therapeutic Effect 

Trong et al. inform [1] (Table 1) that statins seem to be useful medications in the treatment of psoriasis, according to their anti-inflammatory and immunomodulatory features. The pharmacological effects of statins are presented in Figure 2 and summarized in Table 2. Their study shows that statin administration in patients with psoriasis might not only help to control hyperlipidemia but also decrease the Psoriasis Area and Severity Index (PASI) score in psoriatic patients. The therapeutic results from the study are shown in Table 3.

In the second group, which was treated only with calcipotriol/betamethasone dipropionate ointment, no changes in lipid levels during the study were noticed. These results suggested that simvastatin has a positive effect on dyslipidemia. It is difficult to point out which mechanisms of cholesterol modulation caused by statins create beneficial effects on psoriasis lesions. There are many possible explanations, e.g., the inhibition of pro-inflammatory cytokines such as tumor necrosis factor-alpha and interleukin-1 and -6, lowering the level of C-reactive protein, the inhibition of Th1 cytokine receptors on T-cells, the inhibition of the promotion of Th1 to Th2 cells, the downregulation of leukocyte function-associated-1 antigen (LFA-1), the inhibition of leukocyte-endothelial adhesion, extravasation, and natural killer cell activity. All these processes resulted in restrictions of the lymphocyte activation process and an improvement of their influx into the skin, which was proved by a decrease in the PASI score. This study presents also that adding simvastatin (40 mg/d) to calcipotriol/betamethasone dipropionate ointment produced a significantly beneficial effect in psoriatic treatment [1]. Garshick et al. [7] (Table 1) prove that statins can lower cardiovascular (CV) risk in psoriasis (PsO) both through lipid-mediated or a direct effect of statins on the vascular endothelium. In psoriatic patients submitted to the two-week long statin therapy, LDL-C was reduced by ~50% compared to the control group (60 mg/dL vs. 111 mg/dL, *p* < 0.001, respectively). Vascular endothelial inflammation was 60% lower in the statin group vs. the no-treatment group (*p* = 0.02). Change in vascular endothelial inflammation correlated with change in LDL-C (r = 0.53, *p* < 0.05), but not with IL-6 (r = −0.25, *p* = 0.30) or high sensitivity C-reactive protein (hs-CRP) (r = 0.15, *p* = 0.58) [7]. In addition, Naseri et al. [8] (Table 1) show that oral simvastatin enhances the therapeutic effect of topical steroids against psoriasis. The authors divided 30 patients with plaque-type psoriasis into two equal treatment groups (15 patients in each group). The first group received oral simvastatin (40 mg/d) and a topical steroid (50% betamethasone in petrolatum); the second group was given an oral placebo plus the same topical steroid. Both groups were administered medications for eight weeks. In group number 1, the PASI score was 9.51, which declined to 3.38 after 8 weeks of treatment (*p* = 0.001). In group number 2, the PASI score was 5.64, which decreased to 3.98 after 8 weeks of treatment (*p*-value = 0.006). The co-administration of statins and retinoids proved to exert a synergistic anti-psoriatic effect and the additional benefit of normalizing lipid abnormalities [8]. (Table 1) Different studies show that statin intake appears to be associated with a decreased risk of psoriasis recurrence. Those results were based on two analyses. The first study compared subjects who declared that they had psoriasis (n = 356) with their matched controls (n = 1.068). For statin intake, the odds ratio (OR) was 0.64 (0.43–0.97) and *p* = 0.037. The second study, based on multivariate analysis that compared subjects who stated they had psoriasis and were confirmed by a dermatologist or a general practitioner (n = 167), in comparison with matched controls (n = 501). The score for statin intake was an odds ratio (OR) of 0.63 (0.40–1.00) and *p* = 0.051 [9]. Shirinsky et al. [10] (Table 1) report that in their study, 40 mg/d of simvastatin was associated with clinical improvement in psoriasis and was well tolerated. At week 8, there was a statistically significant reduction in the PASI score. The mean reduction in PASI from the beginning of the study to the end of treatment was about 47%. Moreover, two patients have achieved a 50% PASI response, and another two have shown a 75% PASI improvement [10]. A well-known receptor that is a grasping point for statins is liver X receptor alpha (LXR-a). Soodgupta et al. [11] (Table 1) noticed that an increase of 55% in LXR-a. gene expression at the RNA level was observed in the case of use of atorvastatin + 22-R hydroxycholesterol compared to 24% in ascorbic acid + 22-ROH cholesterol (*p* < 0.05). Gene LXR-a possesses anti-inflammatory activity in the skin; apart from that, it may modulate epidermal differentiation and proliferation (by allowing cells to enter the S phase of the cell cycle) and impact the permeability barrier. A decrease in the number of cells in the S-phase of the cell cycle was observed in the treated keratinocytes. The S-phase of the cell cycle was found to be 58.09 percent, 35.27 percent, and 26.32 percent in the keratinocytes derived from the lesioned skin of psoriatic patients treated with atorvastatin + 22-R hydroxycholesterol and ascorbic acid + 22-R hydroxycholesterol, respectively. To sum up, activation of LXR-a could lead to a decrease in the number of cells presented in the S-phase of the cell cycle, thereby decreasing cell proliferation. As a result, LXR-a may play an important role in the therapeutic success of psoriasis [11]. Garshick et al. [12] (Table 1) inform that PsO were given 40 mg of atorvastatin or nothing for two weeks. After atorvastatin treatment, LDL-C was 44% lower in the statin group compared to the no-treatment group [12]. Vasiuk et al. [13] (Table 1) show that six months of therapy with atorvastatin resulted in better treatment results than a treatment scheme without atorvastatin. It was proved by the lowering of PASI from 22.2 to 3.6 compared with a decrease from 22.6 to 17.0 in the control group, *p* < 0.01. In the same study, 47.9% of patients achieved PASI 50% in 3 weeks and 95.8% after 6 months with standard therapy, in comparison to 0% and 13.3%, respectively, in the control group. 81.3% of patients gained a level of PASI of 75% before the 6th month of treatment [13]. Chua et al. [14] (Table 1) notice that after 6 months, mean reductions in PASI scores in the group, where patients were taking atorvastatin, were higher than those of the placebo group. Patients from the atorvastatin group and placebo group showed positive clinical improvement of skin lesions at the end of treatment, but the atorvastatin group had significantly lower total cholesterol and LDL-C levels than the placebo group [14].

#### 3.1.2. Lack of Statins’ Therapeutic Effect on Psoriasis

In contrary to previously mentioned studies, Faghihi et al. [15] (Table 4) noticed that administration of oral atorvastatin 40 mg/day was not associated with therapeutic effect in patients with baseline PASI less than 12, who were along treated with standard topical therapies. The mean baseline PASI scores were 7.42 ± 1.90 and 6.92 ± 1.76 in the atorvastatin and placebo groups, respectively. Significant improvement in psoriasis lesions was observed in both the atorvastatin and placebo groups (*p* < 0.001 for both groups). A 75% progression in PASI score (PASI 75) was achieved in eight patients (40%) in the atorvastatin group and seven patients (35%) in the placebo group [15]. Salman et al. [16] (Table 4) investigated the efficacy and safety of adding simvastatin to narrowband ultraviolet B (NB-UVB) phototherapy in patients with psoriasis. Forty patients with psoriasis who underwent NB-UVB phototherapy were randomly divided into two groups; oral simvastatin was administered to one group, and the other one received a placebo for 12 weeks. Both groups were characterized with a significant decrease in PASI score after 6 and 12 weeks compared to the basic one. Neither the disparities in reducing PASI scores nor the dermatology life quality index (DLQI) between the two groups were prominent at week 6 or week 12. In the simvastatin group, five (25%) patients reached the 75% reduction in PASI (PASI 75) in the 6th week, and six (30%) patients showed the same amount of decrease at the 12th week. In the placebo group, two (10%) and four (23.5%) patients reached PASI 75 at the 6th and 12th weeks, respectively. Looking at PASI 75, no significant differences were seen between the groups either at week six or at week 12 [16]. In another study, Aslam et al. [17] (Table 4) demonstrated that improvement in PASI score in patients who were given simvastatin 40 mg orally once a day for eight weeks was only noticed in 12 patients out of 60 who experienced a reduction in PASI score of >50%. Out of these, 33% of patients had a severe form of the disease, and 67% of cases had moderate plaque psoriasis. Cases of a moderate form of the disease responding to simvastatin were two times more common in comparison to those with severe plaque psoriasis. The authors concluded that the drug seems to be more useful in moderate illnesses. In the remaining 48 (80%) patients, simvastatin wasn’t efficacious. The average percentage reduction in PASI score at the end of treatment was 22.5%, which points out that simvastatin should not be treated as an efficacious drug based on that research. This drug seems to be more useful in moderate psoriasis [17]. 

#### 3.1.3. Statins’ Deteriorating Influence on Psoriasis 

Some authors reported deterioration of the plaque-type of psoriasis during the treatment with statins. Colsman et al. [18] (Table 5) reveal that in four patients who were examined, a small but not significant reduction in PASI was observed during their simvastatin treatment with topical calcipotriol in combination with mometasone or betamethasone valerate. In one patient, the PASI deteriorated temporarily by 50% from 20 to 29, and in another patient, it rose from 7 to 10. Overall, after 12 weeks of treatment, the PASI was almost unchanged from the beginning. During that study, the CRP decreased and the lipid parameters improved [18]. Cozzani et al. [19] (Table 5) inform about 47-year-old psoriatic patient, who was treated with topical therapy for 10 years. During the examination, plaques of psoriasis were observed on the trunk and arms, and scaling lesions were noticed on the plants and palms. The PASI score was 6.8. Laboratory tests revealed hypercholesterolemia. Then, it was decided to administer atorvastatin. Three months later, the skin lesions on his arms got worse and became itchy. On both legs, there were new plaques. The PASI rose from 6.8 to 12.3. Afterwards, atorvastatin was replaced by rosuvastatin. After two months, the lesions improved without any modification of the previous topical medication [19]. Salna et al. [20] (Table 5) report on an 82-year-old patient with comorbidities like gout and hypercholesterolemia who was diagnosed by his physician with the new onset of pruritic, scaly erythematous plaques bilaterally on the extensor surfaces of his arms. The biopsy revealed compact orthokeratosis and significant spongiosis with a perivascular lymphocytic infiltrate, consistent with an eczematous process. No evidence of acanthosis or psoriasiform epidermal hyperplasia. It was his first time having this type of skin lesion. His daily medications were allopurinol 300 mg, aspirin 81 mg, hydrochlorothiazide 25 mg, pravastatin 20 mg, and colchicine 0.6 mg. The man had been taking pravastatin for over 5 years, due to hypercholesterolemia. A few months before the onset of the rash, the dosage of provastatin had been raised from 10 mg to 20 mg daily. After unsuccessful administration of triamcinolone acetonide 0.1% solution and methotrexate, there wasn’t any improvement. Finally, it was decided to rule out pravastatin as an etiologic agent. Statin was switched to cholestyramine due to hypercholesterolemia. As a result, after eight weeks, his cutaneous lesions resolved. This medical history shows that pravastatin can be associated with psoriasis-like eczematous lesions that may be resistant to treatment with steroids and immunosuppressive therapy [20]. Jacobi et al. [21] (Table 5) described the medical history of a 39-year-old man who was dealing with hypercholesterolemia. He had been suffering from intermittent chronic plaque psoriasis since the age of 9 years. When his serum cholesterol level rose to 11.5 mmol/L, he started taking servastatin 100 mg daily. Four days later, he noticed a flare-up of papular psoriasis, which he experienced for the first time ever. Servastatin was discontinued, and he recovered in 6 weeks with additional use of NB-UVB. Afterwards, he began administering simvastatin 40 mg daily. He was obliged to stop it after one week due to the repeated onset of papular psoriasis. This time, the psoriasis was self-limiting within a few days. Next, bezafibrate at a dose of 200 mg daily was initiated, but it had to be stopped after less than a week because his psoriasis had exacerbated again. He was subjected to NB-UVB, with positive effect again. Repeatedly, he started taking pravastatin 10 mg daily. After one week, he had to stop using that drug due to dizziness. Seven days later, the psoriasis flared again; however, it was less severe than on the previous occasions [21]. To sum up, the patient developed recurrent papular exacerbations of his pre-existing psoriasis during the treatment with three different statins (servastatin, simvastatin, and pravastatin) and with fibrate (bezafibrate).

There are many more publications that confirm a positive statin’s effect on psoriasis than articles that describe a neutral or deteriorating statin’s effect on the course of psoriasis. In general, statins may improve psoriatic lesions, but in common practice, we have to be aware of neutral or deteriorating cases in patients suffering from psoriasis. The most important issue is the individual approach to the psoriatic patient and choosing the right and most harmless therapy.

### 3.2. Fibrates’ Influence on Psoriasis with Therapeutic Effects

Fibrates are peroxisome proliferator-activated receptor alpha (PPARα) agonists that regulate lipid metabolism and reduce inflammation through transcriptional regulation. Imamura et al. [2] (Table 6) demonstrated a case where two patients dealing with psoriasis and hypertriglyceridemia were administering 750 mg of clofibrate daily, with successful effects. Additionally, during the clofibrate treatment, hypertriglyceridemia decreased. In the biopsy from the post-treatment plaque, both endothelial swelling in the dermis and capillary proliferation were less prominent. Interestingly, a moderate reduction in the number of lymphocytic cells and a growth in histiocytes were observed. Summing up, clofibrate treatment enhanced triglyceride metabolism and the histological and clinical findings in the psoriatic lesion [2]. Interestingly, Vahlquist et al. [22] (Table 6) inform that after 8 weeks of gemfibrozil treatment, triglycerides in serum were decreased (*p* < 0.01) in patients, who developed a hyperlipidemia, as a side-effect of oral retinoid administration. The total cholesterol level was depleted marginally (*p* < 0.05) during gemfibrozil treatment. Before administering gemfibrozil, patients couldn’t reduce hypertriglyceridemia by using a strict diet or by reducing their dose of retinoids [22].

#### Fibrates’ Deteriorating Influence on Psoriasis

Fisher et al. [23] (Table 7) inform us about a 64-year-old woman who has been suffering from angina for two years. Additionally, she was diagnosed with an elevation of the cholesterol level to 7.19 mmol/L (278 mg/dL) and the LDL-C level to 5.07 mmol/L (196 mg/dL). After two weeks of the gemfibrozil administration, she reported the onset of generalized, papulosquamous, pruritic skin lesions located in the dorsum parts of the arms and hands, with accompanying guttate lesions on the trunk and all extremities. A skin biopsy from her right arm confirmed the presence of psoriasis. Within two weeks, after Balnetar soaks and the clobetasol cream, the psoriasis had almost entirely retreated. Eight weeks later, gemfibrozil therapy was introduced again, and the patient spotted a pronounced exacerbation of the psoriatic lesions, which were localized in the same areas as before. In this connection, she ceased taking gemfibrozil again and took the same topical medications, like the first time, with a totally positive improvement in her condition [23].

Fibrates influence on psoriasis is not as precisely studied as statins impact on patients dealing with psoriasis. Thus far, there are conflicting results, so no conclusions can be drawn yet.

### 3.3. Glitazones’ Influence on Psoriasis

Glitazones have become more and more popular in the treatment of metabolic disorders in recent years. Their mechanism of action involves binding to the peroxisome proliferator-activated receptor (PPAR) gamma, a transcription factor that regulates the expression of specific genes involved in lipid and glucose metabolism in fat cells and other tissues. PPARs primarily function as regulators of FA uptake and oxidation and lipoprotein metabolism. They also act through their influence on cellular proliferation, differentiation, and apoptosis. It became clear that this group of drugs would also interfere with the treatment of psoriasis.

#### 3.3.1. Pioglitazone’s Influence on Psoriasis with Therapeutic Effects

Singh et al. [3] (Table 8) demonstrated 16 patients treated with pioglitazone for 12 weeks who showed major improvements in triglyceride, total cholesterol, and LDL-C levels, as well as in psoriatic symptoms. 81.3% of patients achieved 75% reduction in PASI (*p* < 0.001). The authors suggest that the anti-psoriatic effect of pioglitazone is connected with its anti-inflammatory action [3]. Bongartz et al. [24] (Table 8) present a study where the average reduction in PASI was 38%, with a clinically meaningful PASI 50 response observed in 2 out of the 6 patients who were treated with pioglitazone for 12 weeks [24]. Shafiq et al. [25] (Table 8) demonstrate that psoriasis lesions were diminished in more than 40% of patients who were treated with pioglitazone, for 10 weeks, in comparison to 12.5% of those with placebo. The mean percentage reduction in PASI scores was 47.5%, 41.1%, and 21.6% in the 30 mg pioglitazone, 15 mg pioglitazone, and placebo groups, respectively [25].

#### 3.3.2. Pioglitazone’s Influence on Psoriasis along with Typical Anti-Psoriatic Treatments 

Lajevardi et al. [26] (Table 9) show that pioglitazone therapy boosts the therapeutic effect of methotrexate in plaque-type psoriasis. Forty-four adult patients with psoriasis were included in that study. Patients were divided into two groups: the first group was treated with methotrexate alone (group A), and the second group was treated with methotrexate plus pioglitazone (group B). This study lasted 16 weeks. The reduction in the mean PASI score was 70.3% in group B compared to 60.2% in group A after 16 weeks of therapy. The PASI 75 was achieved in 14 patients (63.6%) in group B compared with 2 patients (9.1%) in group A; the significance was *p* < 0.001. Interestingly, a DLQI decrease of 63.6% was noted in group B, and the reduction of DLQI in group A was 56.9% [26].

Abidi et al. [27] (Table 9) revealed greater PASI reduction in the combination of both methotrexate and pioglitazone after 8 and 12 weeks of treatment in comparison to the therapy with a single drug (methotrexate or pioglitazone). Despite these results, the differences were not significant and need further observation. The results of their study are presented in Table 10. [27]. Mittal et al. [28] (Table 9) note that after 12 weeks of therapy, in the acitretin plus pioglitazone group, the reduction of PASI score was 64.2%, in contrast to the 51.7% PASI decrease in the acitretin plus placebo group [28]. Ghiasi et al. [29] demonstrate that the PASI score, in patients treated for 10 weeks with pioglitazone, was decreased from 20.9 ± 9.8 to 1.8 ± 1.4 (*p* < 0.001), in contrast to the patients who were undertaking phototherapy, where the PASI was reduced from 22 ± 8.5 to 4.4 ± 4. In other terms, PASI was lowered in the pioglitazone group by 83.5% and in the phototherapy group by 56.7% (*p* < 0.05) [29]. In all of the studies beyond this, the extra anti-inflammatory feature of pioglitazone boosted the typical anti-psoriatic and anti-inflammatory action of typical drugs, e.g., acitretin, methotrexate, or phototherapy.

#### 3.3.3. Pioglitazone’s Influence on Psoriasis without Significant Therapeutic Effects

Hafez et al. [30] (Table 11) demonstrate a study in which treatment success (achieving PASI 75) was recorded in just 5 out of 24 (21%) patients who were administered pioglitazone, in comparison to 1 out of 24 (4%) patients receiving a placebo. To sum up, this 10-week long study showed no benefits of taking 30 mg of pioglitazone daily in the clinical symptoms of moderate-to-severe psoriasis. No joint anti-psoriatic therapy was allowed except for vaseline topical daily application and occasional oral antihistamines. Both pioglitazone and placebo were prepared as identical tablets and dispensed in identically labeled and sealed containers of 10 tablets each [30].

#### 3.3.4. Rosiglitazone’s Influence on Psoriasis

Pershadsingh et al. [31] (Table 12) present two patients suffering from psoriasis, both treated with rosiglitazone. During the observation, they had no typical treatment for psoriasis. A 43-year-old nondiabetic man with a 2-year history of plaque psoriasis presented with lesions engaging 10% of his body surface. He started taking rosiglitazone orally, at a dose of 8 mg daily. After 26 weeks of rosiglitazone therapy, the skin lesions like erythema and scaling largely vanished. A second patient, an obese 68-year-old woman with a 5-year history of psoriasis and a 12-year history of type 2 diabetes, was diagnosed with plaque psoriasis occupying over 20% of her body. This patient was administered rosiglitazone orally, in a dose of 4 mg, twice a day, for 24 weeks. In conclusion, a significant improvement in psoriasis was noted. After an additional 26 weeks of taking rosiglitazone, the two large, silvery plaques faded on her lower back, and other lesions appeared over her entire body, including her ears, scalp, and posterior forearms. In conclusion, the therapeutic effect of rosiglitazone on psoriatic lesions was noted after 10 to 26 weeks [31]. In another study, Ellis et al. [32] (Table 12) show that rosiglitazone treatment was not much more effective than placebo treatment in patients who were suffering from moderate-to-severe chronic plaque psoriasis. The rosiglitazone was taken in doses of 2 mg, 4 mg, and 8 mg daily. Study A consisted of 1563 participants, and study B had 1032 participants. Those patients were suffering from moderate-to-severe chronic plaque psoriasis affecting ≥10% body surface area (BSA). In the 26-week study, for patients receiving a placebo, a surprising high response was noticed: PASI 75 for studies A and B achieved 10% (27/261 patients) and 9% (21/245 patients), respectively. In study A, the PASI 75 score for patients taking 2 mg of rosiglitazone was 11% (31/270), for 4 mg of rosiglitazone, it was 12% (34/270), and for 8 mg of rosiglitazone, it was also 12% (36/292). In study B, the PASI 75 score for patients administered 2 mg of rosiglitazone was 11% (31/273) and for 4 mg of rosiglitazone, it was 12% (30/262) [32]. Based on that study, it was found that single treatment of psoriasis with rosiglitazone was not effective in comparison to the effect of placebo therapy.

Reviewing the literature on glitazones effect on psoriasis, it can be concluded that they are effective as a single therapy or in combination with typical anti-psoriatic medicines in patients suffering from psoriasis.

### 3.4. GLP-1 Analogs’ Influence on Psoriasis

GLP-analogs have also shown influence on the lipid serum profile. This new group of drugs targeting diabetic dyslipidemia should potentially have a beneficial effect on psoriatic patients. Apart from its action on body weight and glucose metabolism, GLP-1 can also regulate cholesterol and triglycerides in numerous ways.

#### 3.4.1. Liraglutide’s (a GLP-1 Analog) Influence on Psoriasis with Therapeutic Effects

The GLP-1 receptor is expressed on iNKT cells, while liraglutide may induce a dose-dependent inhibition of iNKT cell cytokine secretion. This action may influence psoriasis severity. Xu et al. [4] (Table 13) inform that liraglutide may effectively improve the psoriatic lesions in patients with type 2 diabetes. After 12 weeks of treatment, the PASI decreased from 15.7 ± 11.8 to 2.2 ± 3.0 (*p* = 0.03), and the DLQI was lowered from 21.8 ± 6 to 4.1 ± 3.9 (*p* = 0.001) [4]. In the presented study, patients did not use typical psoriasis treatment. Lin et al. [33] report that psoriatic lesions in patients dealing with type 2 diabetes improved after the liraglutide treatment. The response rates of PASI 50 and PASI 75 noted in the treatment group were markedly higher than those in the control group (*p* < 0.05). The therapeutic results from the study are shown in Table 14.

After 12 weeks of treatment, both the mean PASI and the mean DLQI scores were markedly reduced in the treatment group in comparison to the control group. In the control group, the PASI score of five patients (38.46%) was reduced by more than 50%, and the score of one patient (7.69%) was depleted by more than 75%. In the treatment group, the PASI score of 10 patients (90.91%) was lowered by more than 50%, and in 8 patients (72.73%), it decreased by more than 75%. Moreover, after 12 weeks of treatment, the mean DLQI score was also much lower in the treatment group in comparison to the control group. A significant liraglutide mechanism can be linked with the inhibition of the expression of inflammatory factors, for instance TNF-α, IL-17, and IL-23, in the psoriatic skin [33]. Faurschou et al. [34] (Table 13) presents a case of 59-years old man with a medical history of hypercholesterolemia, hypertension, acute myocardial infarction and with 15-year long history of plaque psoriasis. Psoriatic lesions were located on the knees, buttocks, elbows, and dorsal parts of the hands and scalp. Afterwards, the treatment with liraglutide started with 0.6 mg once daily, and after one week, the dose rose to 1.2 mg once daily, and after 5 weeks of treatment, to 1.8 mg once daily. Since the start of liraglutide treatment, the patient reported that his psoriatic lesions were reduced, the itching stopped within days, and scaling also decreased. After eight weeks of treatment, spots of normal skin were occurring [34]. Ahern et al. [35] (Table 13) report that 10-week-long therapy with 1.2 mg liraglutide reduced the PASI from 4.8 to 3.0 (*p* = 0.03) and the DLQI from 6.0 to 2.0 (*p* = 0.03) in patients suffering from chronic plaque psoriasis. Two patients (29%) had a bigger than 50% reduction in PASI, and none experienced a larger than 75% reduction in PASI [35].

#### 3.4.2. Liraglutide’s Lack of Influence on Psoriasis 

Faurschou et al. [36] (Table 15) demonstrate that after 8 weeks of liraglutide treatment with changes in PASI, DLQI were not significant in patients with glucose-tolerant obesity and plaque psoriasis compared with placebo. After 8 weeks of treatment, PASI was not changed significantly in the liraglutide group (mean ± standard deviation: −2.6 ± 2.1) compared with the placebo group (−1.3 ± 2.4) (*p* = 0.228). Moreover, no bigger difference in DLQI was observed between the groups [−2.5 ± 4.4 (liraglutide) vs. −3.7 ± 4.8 (placebo); *p* = 0.564]. Additionally, cholesterol levels were lower in the liraglutide group (0.58 ± 0.56) (*p* = 0.045) compared with the placebo group (0.12 ± 0.34) (*p* = 0.045) [36].

#### 3.4.3. Exenatide’s Influence on Psoriasis 

Exenatide has also been demonstrated to have beneficial effects, regarding lipid metabolism, on patients with type 2 diabetes mellitus. However, the potential mechanism remains unclear. Buysschaert et al. [37] (Table 16) inform about major and rapid improvement in psoriatic symptoms in a 61-year-old male patient with diabetes mellitus, thanks to exenatide treatment. Due to the high rates of insulin sensitivity and beta-cell secretion, which were 43% and 29%, respectively, treatment with exenatide was initiated. In the beginning of the study, PASI was 12; after 1 month of the therapy, the patient spotted a huge clinical improvement in psoriatic plaques, which was confirmed at the 12-month follow-up, when PASI was around 3–4. Exenatide therapy had to be discontinued due to administrative reasons, and subsequent withdrawal of exenatide was linked with deterioration of psoriasis; PASI rose to score >10 in just 6 months. After reinstating exenatide therapy, the patient again reported a sudden improvement in his psoriasis PASI score, which decreased to 3.1 [37].

#### 3.4.4. Liraglutide’s and Exenatide’s Influence on Psoriasis

Buysschaert et al. [38] (Table 17) report that administration of GLP-1 analogs (liraglutide and exenatide) ameliorated the severity of clinical chronic plaque psoriasis in patients with type 2 diabetes. The mean PASI lowered from 12.0 ± 5.9 to 9.2 ± 6.4 (*p* = 0.04). Interestingly, histological analysis showed a depletion in the epidermal thickness after GLP-1 analog treatment. Moreover, a correlation between PASI and γδ T-cell amount during therapy was spotted (r = 0.894; *p* = 0.007). IL-17 was also reduced in patients with the highest PASI reductions. This beneficial outcome was connected to a decrease in IL-17 expression and dermal γδ T-cell number [38]. Hogan et al. [39] (Table 17) report about PASI improvement in the following 6 weeks of GLP-1 analog therapy, in obese and diabetic 60-year-old woman who was suffering from refractory and extensive psoriasis since childhood. Before the treatment, she had PASI > 15. She was complaining of very itchy and extensive psoriasis, which was causing marked sleep disruption. Firstly, exenatide was initiated for the treatment of type 2 diabetes. After 2 days, she spotted an improvement in psoriasis severity, which was connected with itch reduction and less sleep disruption. By 2 weeks, the PASI was reduced from 15 to 10.5. The patients had to stop taking exenatide due to nausea after 2 months of treatment. The psoriatic skin lesions quickly began to recur, with PASI reaching 15.3 by the end of 2 weeks. Next, the patient began therapy with liraglutide and reported sudden improvements in the thinning of psoriatic plaques and in the itch. After 3 weeks, the PASI fell from 15.3 to 10.2. After 9 months of continuous liraglutide therapy, psoriatic symptoms remain constant, the PASI score was around 10.5 [39].

#### 3.4.5. Liraglutide’s Influence on Psoriasis along with Typical Anti-Psoriatic Treatments 

Reid et al. [40] (Table 18) inform about a 54-year-old obese man who was dealing with extensive plaque psoriasis. In her past medical history, she was treated with adalimumab, etanercept, efalizumab, ciclosporin, methotrexate, acitretin, fumaric acid esters, narrowband ultraviolet B, and psoralen–ultraviolet. All the drugs mentioned above were ineffective. Then, it was decided to start therapy with 50 mg of acitretin, but that also turned out to be unprofitable. The PASI was 14.2, and the DLQI was 25. Moreover, he did not suffer from diabetes mellitus, but during examination, it turned out that his fasting insulin level was raised to 24.2 mU/L (normal range 2–15 mU/L) and his Homeostasis Model of Assessment-Insulin Resistance (HOMA-IR) was 6.02 (normal value < 2.0). Afterwards, the treatment with liraglutide was initiated at a dose of 0.6 mg daily subcutaneously for 1 week and steadily increased over 6 months to 3 mg, day by day. After 1 year of treatment consisted of liraglutide and acitretin, the DLQI and PASI reduced to 12 and 7.6, respectively. Summing up, the use of GLP-1 analogs, like liraglutide, may offer an effective adjunctive treatment option in psoriasis [40]. Costanzo et al. [41] (Table 18) demonstrate a case of a 73-year-old male patient with obesity, diabetes mellitus, chronic obstructive pulmonary disease, and chronic plaque psoriasis. He suffered from plaque psoriasis for some years; hence, he was treated with topical therapy and adalimumab, but without any success. He was treated for diabetes mellitus with metformin, but that also turned out to be ineffective. At the beginning of that study, the PASI was 33.2 and the DLQI was 26.0. Afterwards, semaglutide (the starting dose was 0.25 mg per week for 4 weeks, then was increased to 0.50 mg per week for 12 weeks, and then to 1 mg per week) was added to metformin. Interestingly, skin lesions from plaque psoriasis improved. After 10 months of therapy, the PASI reduced from 33.2 to 2.6, and DLQI was also lowered from 26.0 to 0. It is well-documented that this anti-psoriatic action of GLP-1 analogs, for example liraglutide, may be associated with the inhibition of macrophage migration, suppressing cell proliferation, impaired inflammation by the activation of adenosine 5′-monophosphate-activated protein kinase, and an increase in circulating invariant natural killer T-cells [41].

Contemporary literature reports that GLP-1 analogs are well-proven and effective therapeutics, either alone or in combination with typical anti-psoriatic treatments, for psoriatic patients.

## 4. Conclusions

Nowadays, we experience a quickly growing interest in the effects of hypolipidemic drugs on the course of psoriasis. The discoveries made in recent years proved the efficacious role of statins, fibrates, glitazones, and analogs of GLP-1 in the additional treatment of psoriasis along with typical anti-psoriatic drugs. In general, the drugs mentioned beyond, through various mechanisms, lead to a more or less marked alleviation of the skin lesion in patients suffering from psoriasis. It can be noted by the reduction of PASI, DLQI, and various pro-inflammatory cytokines. On the other side, the cases of the psoriasis exacerbation were described, after taking above-mentioned drugs. In that case, the course of treatment should be monitored more frequently. Although this subject was broadly researched, large scale, cohort studies are needed to evaluate full effectiveness of hypolipidemic drugs on psoriasis. These studies may help us discover new treatment modalities for psoriasis.

## Figures and Tables

**Figure 1 metabolites-13-00493-f001:**
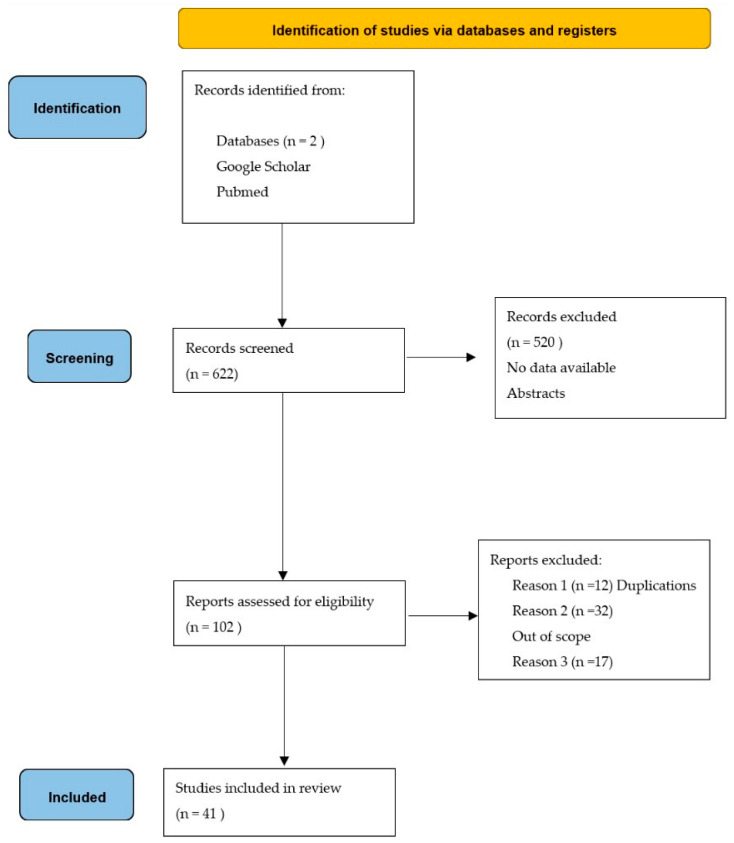
The search process.

**Figure 2 metabolites-13-00493-f002:**
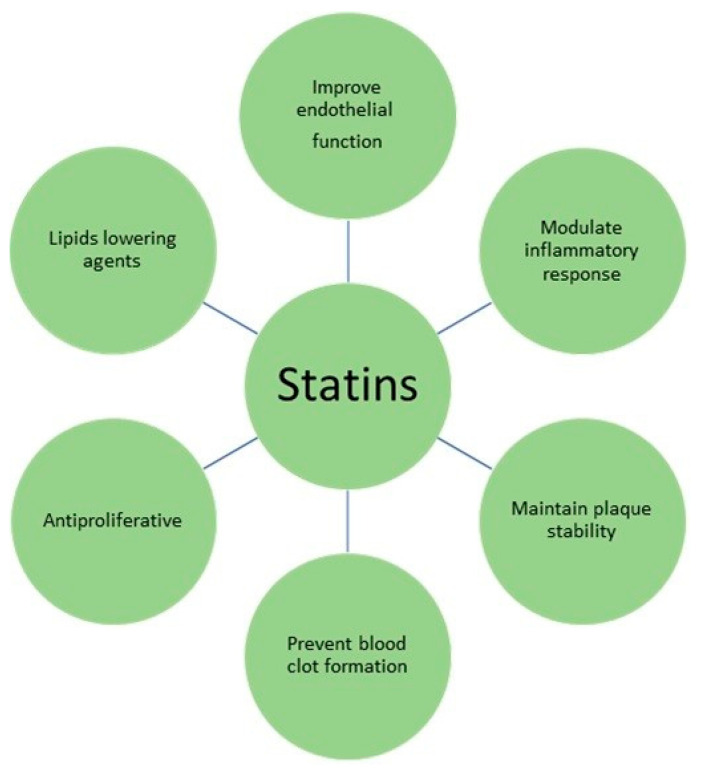
Major pharmacological effects of statins.

**Table 1 metabolites-13-00493-t001:** Summary of the studies on statins’ influence on psoriasis with therapeutic effects.

Author	Year	Population	Key Observations
Statins’ Influence on Psoriasis with Therapeutic Effects			
Trong et al. [1]	2019	n1—128 patients n2—128 patients	Simvastatin might play a role in controlling hyperlipidemia, and in turn decrease the PASI scores in psoriatic patients.
Garshick et al. [7]	2020	n1—10 patients n2—15 patients	Statins can lower cardiovascular risk in psoriasis through lipid-mediated or direct effect of statins on the vascular endothelium.
Naseri et al. [8]	2010	n1—15 patients n2—patients	Oral simvastatin enhances the therapeutic effects of topical steroids against psoriasis.
Wolkenstein et al. [9]	2009	First study n1—1068 patients n2—356 patients Second study n1—501 patients n2—167 patients	Statins appeared to be associated with a decreased risk of psoriasis.
Shirinsky et al. [10]	2007	n—7 patients	Simvastatin at a dosage of 40 mg/d was associated with clinical improvements for psoriasis and was well tolerated.
Soodgupta et al. [11]	2014	n—psoriatic patients (from 18 to 40 years)	LXR may play a big role in therapeutic importance for psoriasis.
Garschick et al. [12]	2022	n1—10 patients n2—20 patients	After the treatment, LDL-C was 44% lower in the statin group compared to the no-treatment group
Vasiuk et al. [13]	2010	n1—15 patients n2—48 patients	Six months of therapy with atorvastatin resulted in a significant lowering of the PASI compared to the control group.
Chua et al. [14]	2017	n1—6 patients n2—8 patients	After 6 months, mean reductions in PASI scores of the group where patients took atorvastatin were higher than those of the placebo group.

Abbreviations: n—general group; n1—control group, n2—treatment group.

**Table 2 metabolites-13-00493-t002:** Specific information about the pharmacological effects of statins [1,7,8,9,10,11,12,13,14,15,16,17,18,19,20,21].

Lipids lowering agents	Statins inhibit 2-hydroxyl-methyl-glutaryl coenzyme A (HMG-CoA) reductase, the rate-limiting step in cholesterol synthesis. This process leads to a reduced intracellular cholesterol concentration and the removal of LDL-C from the circulation.
Improvement of endothelial function	Statins improve flow-mediated dilatation. Statins reduce caveolin 1 levels, decreasing its inhibitory effects on nitric oxide synthesis, which has been shown to inhibit several components of the atherogenic process.
Modulation of inflammatory response	Statins reduce C-reactive protein (CRP) levels. Statins inhibit MHC-II expression on endothelial cells and monocyte-macrophages via inhibition of the promotor IV of the transactivator CIITA, and thereby repress MHC-II-mediated T-cell activation. In addition, statins have been shown to decrease CD40 expression and CD40-related activation of vascular cells. Moreover, statins might decrease the activity of activator protein-1 (AP-1), which regulate genes responsible for metalloproteinases (MMPs), cytokines, chemokines, adhesion molecules, and inducible nitric oxide synthase (iNOS).
Maintenance of plaque stability	Statins reduce the in vitro cholesterol accumulation in macrophages and the expression of matrix metalloproteinases, which subsequently leads to plaque stability.
Prevention of blood clot formation	Statins’ actions were associated with reduced rates of prothrombin activation, factor Va generation, fibrinogen cleavage, factor XIII activation, and increased rates of factor Va inactivation.
Antiproliferative function	Treatment with statins decreased PDGF-induced Rb hyperphosphorylation and cyclin-dependent kinases (cdk)-2, -4, and -6 activities. This correlated with increased levels of the Cdk inhibitor( i.e., p27^Kip1^) without concomitant changes in p16^INK4^, p21^Waf1^, or p53 levels. These findings indicate that statins inhibit vascular smooth muscle cell proliferation by arresting the cell cycle between G1/S phase transitions.

**Table 3 metabolites-13-00493-t003:** Results of treatments with statins reported by Trong et al. [1].

Patient’s Group	Before Treatment	*p*	4th Week	*p*	8th Week	*p*
Cholesterol concentration	5.45 ± 1.21 mm/L	<0.001	4.20 ± 0.82 mm/L	<0.001	4.18 ± 0.72 mm/L	<0.001
Triglyceride level	1.86 ± 1.17 mm/L	0.07	1.32 ± 0.84 mm/L	0.07	1.26 ± 0.65 mm/L	<0.005
LDL-c level	3.18 ± 0.7 mm/L	<0.001	2.31 ± 0.80 mm/L	<0.001	2.26 ± 0.7 mm/L	<0.001
HDL levels	1.33 ± 0.31 mm/L	<0.001	1.29 ± 0.24 mm/L		1.35 ± 0.24 mm/L	
PASI	12.8 ± 5.87		8.58 ± 5.62	<0.01	4.17 ± 3.81	<0.001

**Table 4 metabolites-13-00493-t004:** Summary of the studies on statins’ influence on psoriasis without significant therapeutic effects.

Author	Year	Population	Key Observations
Statins’ Influence on Psoriasis without Significant Therapeutic Effects			
Faghihi et al. [15]	2011	n1—20 patients n2—20 patients	Oral atorvastatin (40 mg/day) was not associated with therapeutic benefits when given to patients with baseline PASI scores less than 12, who were also treated with standard topical therapies.
Salman et al. [16]	2021	n1—20 patients n2—20 patients	Both groups were characterized with a significant decrease in the PASI score after 6 and 12 weeks, compared to the basic one.
Aslam et al. [17]	2013	n—60 patients	The improvement in PASI scores in patients who were given simvastatin was only noticed in 12 patients out of 60; they experienced a reduction in the PASI score of >50%.

Abbreviations: n—general group; n1—control group, n2—treatment group.

**Table 5 metabolites-13-00493-t005:** Summary of the studies on statins’ deteriorating influence on psoriasis.

Author	Year	Population	Key Observations
Statins’ Deteriorating Influence on Psoriasis			
Colsman et al. [18]	2010	n—5 patients	In one patient the PASI deteriorated temporarily by 50% from 20 to 29, and in another patient it raised from 7 to 10.
Cozzani et al. [19]	2009	n—1 patient	After initiation with atorvastatin, the psoriatic lesions on his arms worsened and became itchy. New plaques formed on both of his legs. The PASI rose from 6.8 to 12.3.
Salna et al. [20]	2017	n—1 patient	Pravastatin can be associated with psoriasis-like eczematous lesions which may be resistant to the treatment with steroids or immunosuppressive therapies.
Jacobi et al. [21]	2003	n—1 patient	Servastatin may flare-up papular psoriasis.

Abbreviations: n—general group.

**Table 6 metabolites-13-00493-t006:** Summary of the studies on fibrate’s influence on psoriasis with therapeutic effects.

Author	Year	Population	Key Observation
Fibrate’s Influence on Psoriasis with Therapeutic Effects			
Imamura et al. [2]	1991	n—2 patients	Clofibrate treatment improved triglyceride levels and the histological and clinical findings in psoriatic lesions.
Vahlquist et al. [22]	1995	n—14 patients	Gemfibrozil appears useful in patients prone to retinoid-induced hyperlipidemia that is unresponsive to dietary treatments and acitretin dose reductions.

Abbreviations: n—general group.

**Table 7 metabolites-13-00493-t007:** Summary of the studies on fibrate’s deteriorating influence on psoriasis.

Author	Year	Population	Key Observations
Fibrate’s Deteriorating Influence on Psoriasis			
Fisher et al. [23]	1988	n—1 patient	Gemfibrozil administration may cause papulosquamous skin lesions.

Abbreviations: n—general group.

**Table 8 metabolites-13-00493-t008:** Summary of the studies on pioglitazone’s influence on psoriasis with therapeutic effects.

Author	Year	Population	Key Observations
Pioglitazone’s Influence on Psoriasis with Therapeutic Effects			
Singh et al. [3]	2016	n1—23 patients n2—16 patients n3—21 patients	Pioglitazone treatment of 12 weeks showed major improvements in triglycerides, total cholesterol, LDL-C levels, and psoriatic symptoms.
Bongartz et al. [24]	2004	n—10 patients	In people who were taking pioglitazone, the average reduction in PASI was 38%.
Shafiq et al. [25]	2005	n1—25 patients n2—21 patients n3—24 patients	Two-thirds of patients with plaque psoriasis seem to improve with the use of pioglitazone.

Abbreviations: n—general group; n1—control group, n2,3—treatment group.

**Table 9 metabolites-13-00493-t009:** Summary of the studies on pioglitazone’s influence on psoriasis along with typical anti- psoriatic treatments.

Author	Year	Population	Key Observations
Pioglitazone’s Influence on Psoriasis Along with Typical Anti-Psoriatic Treatments			
Lajevardi et al. [26]	2014	n1—22 patients n2—22 patients	The combination therapy of pioglitazone and methotrexate has a major therapeutic effect on the severity of psoriasis than the therapy with a single factor.
Abidi et al. [27]	2020	n1—30 patients n2—30 patients n3—30 patients	Combination of methotrexate and pioglitazone proved superior in efficacy.
Mittal et al. [28]	2009	n1—22 patients n2—19 patients	The percentage of reduction in the PASI score was more major in the acitretin plus pioglitazone group, than in the acitretin plus placebo group.
Ghiasi et al. [29]	2018	n1—30 patients n2—30 patients	Pioglitazone can vastly enhance the effectiveness of phototherapy in patients with plaque psoriasis.

Abbreviations: n—general group; n1—control group, n2,3—treatment groups.

**Table 10 metabolites-13-00493-t010:** Results of treatment, measured by PASI, with statins reported by Abidi et al. [27].

Weeks	Group A (Methotrexate 7.5 mg/Week for 12 Weeks)	Group B (Pioglitazone 15 mg Tablets Once Daily for 12 Weeks)	Group C (Methotrexate 7.5 mg/Week and Pioglitazone 15 mg/Day)	
	PASI	PASI	PASI	*p*
0	17.68 ± 1.103	17.73 ± 1.203	18.12 ± 1.419	0.275
4	11.86 ± 1.062	11.75 ± 1.486	11.87 ± 1.317	0.929
8	6.717 ± 1.026	6.723 ± 1.428	5.807 ± 1.069	0.004
12	3.797 ± 0.6185	4.540 ± 1.467	3.063 ± 0.6178	<0.0001

**Table 11 metabolites-13-00493-t011:** Summary of the studies on pioglitazone’s influence on psoriasis without significant therapeutic effects.

Author	Year	Population	Key Observations
Pioglitazone’s Influence on Psoriasis without Significant Therapeutic Effects			
Hafez et al. [30]	2015	n1—24 patients n2—24 patients	This study revealed that pioglitazone (30 mg daily) produced no therapeutic effect on the moderate-to-severe psoriasis.

Abbreviations: n—general group; n1—control group, n2—treatment group.

**Table 12 metabolites-13-00493-t012:** Summary of the studies on rosiglitazone’s influence on psoriasis.

Author	Year	Population	Key Observations
Rosiglitazone’s Influence on Psoriasis			
Pershadsingh et al. [31]	2005	n—2 patients	Rosiglitazone may be useful in treating psoriasis vulgaris.
Ellis et al. [32]	2007	n—2595 patients	Rosiglitazone treatment was not more effective than the placebo treatment in patients who were suffering from moderate-to-severe chronic plaque psoriasis

Abbreviations: n—general group; n1—control group, n2—treatment group.

**Table 13 metabolites-13-00493-t013:** Summary of the studies on liraglutide’s influence on psoriasis with therapeutic effects.

Author	Year	Population	Key Observations
Liraglutide’s Influence on Psoriasis with Therapeutic Effects			
Xu et al. [4]	2019	n—7 patients	Liraglutide can improve psoriatic skin lesions in patients with type 2 diabetes, especially in cases of extremely severe psoriasis.
Lin et al. [33]	2020	n1—13 patients n2—12 patients	Skin lesions in psoriatic patients with type 2 diabetes were significantly improved after the treatment with liraglutide. This may be related to the inhibition of the expression of inflammatory factors, such as IL-23, IL-17, and TNF-a.
Faurschou et al. [34]	2011	n—1 patient	Liraglutide markedly improved psoriasis in a patient treated for type 2 diabetes.
Ahern et al. [35]	2012	n—7 patients	Liraglutide therapy improves psoriasis severity, increases the circulating iNKT cell number, and modulates monocyte and cytokine secretions.

Abbreviations: n—general group; n1—control group, n2—treatment group, iNKT—invariant natural killer-T.

**Table 14 metabolites-13-00493-t014:** Results of treatment with statins, measured by PASI, reported by Lin et al. [33].

Week	Control Group			Treatment Group		
	PASI	DLQI	*p*	PASI	DLQI	*p*
0	13.57 ± 5.49	18.23 ± 5.17	<0.05	14.02 ± 10.67	22.00 ± 5.85	<0.05
12	7.42 ± 3.91	9.69 ± 4.59	<0.05	2.40 ± 2.71	3.82 ± 3.60	<0.05

**Table 15 metabolites-13-00493-t015:** Summary of the studies on liraglutide’s lack of influence on psoriasis.

Author	Year	Population	Key Observations
Liraglutide’s Lack of Influence on Psoriasis			
Faurschou et al. [36]	2014	n1—9 patients n2—11 patients	Liraglutide treatment of 8 weeks did not significantly change PASI and DLQI scores in a group of patients with plaque psoriasis, compared to the placebo group.

Abbreviations: n—general group; n1—control group, n2—treatment group.

**Table 16 metabolites-13-00493-t016:** Summary of the studies on exenatide’s influence on psoriasis.

Author	Year	Population	Key Observations
Exenatide’s Influence on Psoriasis			
Buysschaert et al. [37]	2012	n—1 patient	There was a major and rapid improvement in psoriasis in our patients with diabetes mellitus following the exenatide treatment.

Abbreviations: n—general group; n1—control group, n2—treatment group.

**Table 17 metabolites-13-00493-t017:** Summary of the studies on liraglutide’s and exenatide’s influence on psoriasis.

Author	Year	Population	Key Observations
Liraglutide’s and Exenatide’s Influence on Psoriasis			
Buysschaert et al. [38]	2014	n—7 patients	The administration of an exenatide/liraglutide improved the severity of clinical psoriasis in patients with type 2 diabetes. This favorable outcome was associated with a decrease in dermal cd T-cell numbers and the IL-17 expression.
Hogan et al. [39]	2011	n—2 patients	The PASI improved in both patients following 6 weeks of liraglutide and exenatide therapy.

Abbreviations: n—general group, n1—control group, n2—treatment group.

**Table 18 metabolites-13-00493-t018:** Summary of the studies on liraglutide’s influence on psoriasis along with typical anti-psoriatic treatments.

Author	Year	Population	Key Observations
Liraglutide’s Influence on Psoriasis along with Typical Anti-Psoriatic Treatments			
Reid et al. [40]	2013	n—1 patient	Liraglutide may be an effective adjunctive treatment option against psoriasis.
Costanzo et al. [41]	2021	n—1 patient	During semaglutide with typical anti-psoriatic treatment, a rapid improvement of severe psoriasis lesions was observed.

Abbreviations: n—general group.

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
