# Peer review of "Effects of Hypolipidemic Drugs on Psoriasis"

_metabolites, 2023, doi:10.3390/metabo13040493_

Round 1

Reviewer 1 Report

The review gives an interesting overview about how hypolipemic drugs can affects proriasis course.

They presented different hypolipemic drugs with different mechanisms of actions mentioning both studies proving the efficacy and studies in which they did not see any beneficial effects. The topic seems to be entire covered. However, I reconsidered the paper after major revisions. Following I listed my comments.

Major comments:

- LINE 134-148: this part, in my opinion, is very difficult follow the results and the comments to data. My suggestion is to rephrase and replace the data with a table to help the reader.

- I suggest including tables to present the results to help the readers to better understand and to give more fluently the paper (i.e. chapters 3.1, 3.4.1).

- LINE 175:  they wrote “The authors divided 30 patients with plaque type psoriasis into two equal treatment groups”, The meaning of this sentence is that the 30 patients were divided into two groups, 15 patients per group. In my opinion it is better to specify that.

- LINE 367: the authors mentioned a study conducted by Abibi et al.  in which it was demonstrated a higher efficacy of methotrexate plus pioglixatone in comparison with methotrexate alone and pioglixatone alone, however the PASI score reduction is more or less the same. How can the authors explain these results? In my opinion it is better to argue this data, to clarify the importance of PASI for treatment decision.

- Line 41-43 “The researcher with his team, demonstrate that during clofibrate treatment on daily bases, show significant improvement both in dyslipidemia abnormalities and from the post-treatment skin samples”. This sentence is confused, I suggest to revise.

- I suggest to give a general comment for drug effects as a general conclusion.

- Overall, my suggestion is mentioning the table N in the main test, since it is very difficult to searching for the data mentioned in the test to the corresponding table.

Minor comments:

In the paper I found a lot of grammatical and types  errors that I suggest to revise. Following I listed some of them:

- Line10 Disturbances is repeated and written wrong

- Line 18 levels, not leveles

- Line 22 the decrease of…..and the increase of…, not decreases and increase

- Line 25 Treatments, not treatment

- Line 26 includes, not inlcuds

- Line 27 until of December, without the

- Line 35 characterized by, not with

- Line 37 On one hand, not on one the hand

- Line 38 to affect the, not to affect on

- Line 46 the acronym should be written entirely

- Line 52 Originally, instead of original

- Line 140 Starting from here, brackets are used improperly, once opened are not closed.

- Line 154 difficult, not diffuclt

- Line 162 lymphocyte, not limphocyte

- Line 163 cutaneous which is an adjective should be replaced by a substantive

- Line 169 in my opinion I suggest to replace “no treatment group” with “control group”

- Line 184 analisis/ analysis and those results was/were

- Line 241 Neither or/nor

- Line 261 I suggest to re-phrase “it was observed a small reduction, but not significant reduction in PASI”, the word “reduction” is redundant.

- Line 305 “Fibrates are peroxisome proliferator-activated receptor alpha (PPARα) agonists that regulate lipid metabolism and reduce inflammation through transcriptional regulation.” It is repeated two times.

- Line 352 was spotted should be replaced by spotted

- Line 370, 371, 372, 373, 374 I suggest to replace “in” preposition with “at”

- Line 402 I suggest replacing “in results” with “in conclusion” or “summary”

- Line 458 I suggest replacing “,” with “and”

- Line 459 PASI was, not were

- Line 473 I suggest rephrasing “when a PASI score” with “when PASI score”

- Line 485 I suggest rephrasing “IL-17 was also reduced in the with the highest PASI reductions.”

- Line 493 the first with should be replaced by which

- Line 503 In her past medical history she was, not he.

Author Response

Reviewer 1

The review gives an interesting overview about how hypolipemic drugs can affects proriasis course.

They presented different hypolipemic drugs with different mechanisms of actions mentioning both studies proving the efficacy and studies in which they did not see any beneficial effects. The topic seems to be entire covered. However, I reconsidered the paper after major revisions. Following I listed my comments.

Thank you for the time spent on the revision of our paper and for the constructive remarks. We have made the corrections that you have suggested and hope that with your help we managed to improve our manuscript.

                                                           Major comments

1.LINE 134-148: this part, in my opinion, is very difficult follow the results and the comments to data. My suggestion is to rephrase and replace the data with a table to help the reader.

Thank you very much for you suggestion. According to your suggestion, I put data into a table to help the reader.

  1. I suggest including tables to present the results to help the readers to better understand and to give more fluently the paper (i.e. chapters 3.1, 3.4.1).

Thank you very much for you suggestion. According to your suggestion, I included tables  with data to better present it for readers, and what is more I gave more fluently the paper, by correcting name of chapters.

  1. LINE 175:  they wrote “The authors divided 30 patients with plaque type psoriasis into two equal treatment groups”, The meaning of this sentence is that the 30 patients were divided into two groups, 15 patients per group. In my opinion it is better to specify that.

Thank you very much for your remark. According to your suggestion, I specified that information.

  1. LINE 367: the authors mentioned a study conducted by Abibi et al.  in which it was demonstrated a higher efficacy of methotrexate plus pioglixatone in comparison with methotrexate alone and pioglixatone alone, however the PASI score reduction is more or less the same. How can the authors explain these results? In my opinion it is better to argue this data, to clarify the importance of PASI for treatment decision.

Thank you very much for your remark. The authors explained it that the difference in PASI reduction, after combination of methotrexate and pioglitazone, was more visible after 8 and 12 weeks of treatment, not after 4 weeks in comparison to the therapies with single drugs (methotrexate or pioglitazone). Moreover, I put PASI scores after 8 weeks of treatment in the manuscript to present in more clear way for readers, the depletions of PASI scores during treatment.

  1. Line 41-43 “The researcher with his team, demonstrate that during clofibrate treatment on daily bases, show significant improvement both in dyslipidemia abnormalities and from the post-treatment skin samples”. This sentence is confused, I suggest to revise.

Thank you very much for your remark. According to your suggestion, I revised those sentence into more clear one.

  1. I suggest to give a general comment for drug effects as a general conclusion.

Thank you very much for your remark. According to your suggestion, I gave a general comment for drug effects as a general conclusion.

  1. Overall, my suggestion is mentioning the table N in the main test, since it is very difficult to searching for the data mentioned in the test to the corresponding table.

Thank you very much for your remark. According to your suggestion, I mentioned tables in the main text.

Minor comments:

1.Line10 Disturbances is repeated and written wrong.

Thank you very much for your remark. According to your suggestion, I corrected it.

  1. Line 18 levels, not leveles.

Thank you very much for your remark. According to your suggestion, I corrected it.

  1. Line 22 the decrease of…..and the increase of…, not decreases and increase

Thank you very much for your remark. According to your suggestion, I corrected it.

  1. Line 25 Treatments, not treatment.

Thank you very much for your remark. According to your suggestion, I corrected it.

  1. Line 26 includes, not inlcuds.

Thank you very much for your remark. According to your suggestion, I corrected it.

  1. Line 27 until of December, without the.

Thank you very much for your remark. According to your suggestion, I corrected it.

  1. Line 35 characterized by, not with.

Thank you very much for your remark. According to your suggestion, I corrected it.

  1. Line 37 On one hand, not on one the hand.

Thank you very much for your remark. According to your suggestion, I corrected it.

  1. Line 38 to affect the, not to affect on.

Thank you very much for your remark. According to your suggestion, I corrected it.

  1. Line 46 the acronym should be written entirely

Thank you very much for your remark. According to your suggestion, I corrected it.

  1. Line 52 Originally, instead of original

Thank you very much for your remark. According to your suggestion, I corrected it.

  1. Line 140 Starting from here, brackets are used improperly, once opened are not closed.

Thank you very much for your remark. According to your suggestion, I corrected it.

  1. Line 154 difficult, not diffuclt.

Thank you very much for your remark. According to your suggestion, I corrected it.

  1. 14. Line 162 lymphocyte, not limphocyte

Thank you very much for your remark. According to your suggestion, I corrected it.

  1. Line 163 cutaneous which is an adjective should be replaced by a substantive

Thank you very much for your remark. According to your suggestion, I corrected it.

  1. Line 169 in my opinion I suggest to replace “no treatment group” with “control group”

Thank you very much for your remark. According to your suggestion, I corrected it.

  1. Line 184 analisis/ analysis and those results was/were.

Thank you very much for your remark. According to your suggestion, I corrected it.

  1. Line 241 Neither or/nor

Thank you very much for your remark. According to your suggestion, I corrected it.

  1. Line 261 I suggest to re-phrase “it was observed a small reduction, but not significant reduction in PASI”, the word “reduction” is redundant.

Thank you very much for your remark. According to your suggestion, I corrected it.

  1. Line 305 “Fibrates are peroxisome proliferator-activated receptor alpha (PPARα) agonists that regulate lipid metabolism and reduce inflammation through transcriptional regulation.” It is repeated two times.

Thank you very much for your remark. According to your suggestion, I corrected it.

  1. Line 352 was spotted should be replaced by spotted.

Thank you very much for your remark. According to your suggestion, I changed it into observed.

  1. Line 370, 371, 372, 373, 374 I suggest to replace “in” preposition with “at”.

Thank you very much for your remark. According to your suggestion, I corrected it.

  1. Line 402 I suggest replacing “in results” with “in conclusion” or “summary”.

Thank you very much for your remark. According to your suggestion, I corrected it.

  1. Line 458 I suggest replacing “,” with “and”.

Thank you very much for your remark. According to your suggestion, I corrected it.

  1. Line 458 I suggest replacing “,” with “and”.

Thank you very much for your remark. According to your suggestion, I corrected it.

  1. Line 459 PASI was, not were.

Thank you very much for your remark. According to your suggestion, I corrected it.

  1. Line 473 I suggest rephrasing “when a PASI score” with “when PASI score”

Thank you very much for your remark. According to your suggestion, I corrected it.

  1. Line 485 I suggest rephrasing “IL-17 was also reduced in the with the highest PASI reductions.”

Thank you very much for your remark. According to your suggestion, I corrected it.

  1. Line 493 the first with should be replaced by which.

Thank you very much for your remark. According to your suggestion, I corrected it.

  1. Line 503 In her past medical history she was, not he.

Thank you very much for your remark. According to your suggestion, I corrected it.

Reviewer 2 Report

Manuscript written by Mateusz et al. is a valuable review work, they reviewed several papers that related to the effect of different hypolipemic treatment on the course of psoriasis. However, several aspects should be taken into account. Please see below:

1.      First figure in the paper (PRISMA diagram) was divided into two pages, it would be better if the author could put them in the same page.

2.      It would be better if the author could add a new table for the Fig 1 to provided more specific information about the pharmacological effects of statins.

3.      In the “Materials and Methods” part, the author could add a time frame for the google scholar like they showed in the PubMed (1992-present).

4.      The author mentioned that they selected these papers which were published after 1992. However, this paper entitled “Clofibrate treatment of psoriasis with hypertriglycemia-clinical, histological and laboratory analysis” was published in 1991.

5.      There are some minor language errors. The authors should be revised the manuscript with an English language editor to make it more readable.

Author Response

Reviewer 2

Manuscript written by Mateusz et al. is a valuable review work, they reviewed several papers that related to the effect of different hypolipemic treatment on the course of psoriasis. However, several aspects should be taken into account. Please see below:

Thank you for the time spent on the revision of our paper and for the constructive remarks. We have made the corrections that you have suggested and hope that with your help we managed to improve our manuscript.

  1. First figure in the paper (PRISMA diagram) was divided into two pages, it would be better if the author could put them in the same page.

Thank you for your remark. According to your suggestion, I put that PRISMA diagram in one page.  

  1. It would be better if the author could add a new table for the Fig 1 to provided more specific information about the pharmacological effects of statins.

Thank you for your suggestion. According to your advice, I provide more specific  information about the pharmacological effects of statins.

  1. In the “Materials and Methods” part, the author could add a time frame for the google scholar like they showed in the PubMed (1992-present).

Thank you for your suggestion. According to you suggestion, I add a time frame for the google scholar like it was shown in the Pubmed.

  1. The author mentioned that they selected these papers which were published after 1992. However, this paper entitled “Clofibrate treatment of psoriasis with hypertriglycemia-clinical, histological and laboratory analysis” was published in 1991.

Thank you for your remark. I have made that mistake by accident. According to you suggestion, I changed the date into correct one.

  1. There are some minor language errors. The authors should be revised the manuscript with an English language editor to make it more readable.

Thank you for your remark. According to your suggestion, we have corrected minor language errors to be more readable.

Round 2

Reviewer 1 Report

Thanks to the authors for the revisions, I'm satisfied and I agree for the pubblication.

Author Response

Thank you very much for your time, for all your remarks and for checking my manuscript. 

Reviewer 2 Report

The authors addressed most of my concerns; now the paper is more readable.

Author Response

(The authors gave the same response as above.)
